# Impact of Two Different Recruitment Procedures (Random vs. Volunteer Selection) on the Results of Seroepidemiological Study (SARS-CoV-2)

**DOI:** 10.3390/ijerph18189928

**Published:** 2021-09-21

**Authors:** Maksymilian Gajda, Małgorzata Kowalska, Jan Eugeniusz Zejda

**Affiliations:** Department of Epidemiology, Faculty of Medical Sciences in Katowice, Medical University of Silesia, 40-752 Katowice, Poland; mkowalska@sum.edu.pl (M.K.); jzejda@sum.edu.pl (J.E.Z.)

**Keywords:** SARS-CoV-2, seroepidemiological study, recruitment, randomization, self-selection

## Abstract

The proper recruitment of subjects for population-based epidemiological studies is critical to the external validity of the studies and, above all, to the sound and correct interpretation of the findings. Since 2020, the novel coronavirus SARS-CoV-2 pandemic has been a new factor that has been, additionally, hindering studies. Therefore, the aim of our study is to compare demographic, socio-economic, health-related characteristics and the frequency of SARS-CoV-2 infection occurrence among the randomly selected group and the group composed of volunteers. We compare two groups of participants from the cross-sectional study assessing the seroprevalence of SARS-CoV-2 coronavirus, which was conducted in autumn 2020, in three cities of the Silesian Voivodeship in Poland. The first group consisted of a randomly selected, nationally representative, age-stratified sample of subjects (1167 participants, “RG” group) and was recruited using personal invitation letters and postal addresses obtained from a national registry. The second group (4321 volunteers, “VG” group) included those who expressed their willingness to participate in response to an advertisement published in the media. Compared with RG subjects, volunteers were more often females, younger and professionally active, more often had a history of contact with a COVID-19 patient, post-contact nasopharyngeal swab, fewer comorbidities, as well as declared the occurrence of symptoms that might suggest infection with SARS-CoV-2. Additionally, in the VG group the percentage of positive IgG results and tuberculosis vaccination were higher. The findings of the study confirm that surveys limited to volunteers are biased. The presence of the bias may seriously affect and distort inference and make the generalizability of the results more than questionable. Although effective control over selection bias in surveys, including volunteers, is virtually impossible, its impact on the survey results is impossible to predict. However, whenever possible, such surveys could include a small component of a random sample to assess the presence and potential effects of selection bias.

## 1. Introduction

The proper recruitment of subjects for population-based epidemiological studies is critical to the external validity of the studies and, above all, to the sound and correct interpretation of the findings. The problem poses a challenge in any selection procedure that aims at the representativeness of the study group. However, it is particularly important in large surveys that rely on convenient face-to-face or telephone interviews. The latter methods differ in terms of application difficulties and cost-effectiveness [1,2]. The common dilemma of this form of research stems from the fact that such surveys are usually limited to volunteers and, thus, are affected by selection bias. Another concern is related to a usually large number of refusals, making it difficult to obtain an appropriate sample, also in terms of its size [3]. Thus, novel recruitment strategies are sought, also with the use of new forms of communication (e.g., the internet). Voluntary recruitment is known to be an important source of non-response bias and volunteer bias [1,4]. However, the exact dimension of the bias and its consequences are seldom reported.

There are many examples of different seroprevalence studies involving volunteers. The study conducted in Italy among adults over 65 years of age showed an overall seroprevalence of anti-SARS-CoV-2 antibodies of 4.7% [5]. In the study performed in Massachusetts at the end of the first wave of COVID-19 pandemic, the incidence of infection was lower in the representative sample that in volunteers: 1.85% vs. 3.29%, respectively [6]. The occurrence of SARS-CoV-2 infection in employees of a large teaching hospital in England examined between May and July 2020 was estimated to be 17.4% [7]. The Italian cohort showed a seroprevalence of 14.4% in a period from March to June 2020 [8]. During the first months of the COVID-19 pandemic, Poland reported a lower incidence of confirmed cases compared to other European countries [9]. The seroprevalence in the population of Poznan metropolitan area in Poland was 1.67% (sample collection between the end of July and the end of September 2020), finally dropping to 0.93% after immunoblotting verification [9]. As in other countries, Polish health care workers were also tested. Samples collected from the staff working at the Children’s Memorial Health Institute in Warsaw (Poland) between July and August 2020 resulted in a seroprevalence of 0.85% [10]. The IgG seropositivity of asymptomatic healthcare workers from southern Poland varied between 1.2% and 10% (July/August 2020) [11]. In our recent seroepidemiological study on SARS-CoV-2 infection, we examined randomly selected subjects using questionnaires and immunological tests [12]. Our project also made possible applying the same research tools in a large group of volunteers, recruited from the same study area and examined in the same study period. It allowed us to explore, in a real-life setting, potential differences between both groups, in terms of information provided by questionnaires and immunological tests. The objective of our study was to compare demographic, socio-economic, health-related characteristics, and the frequency of SARS-CoV-2 infection occurrence between the randomly selected group and the group composed of volunteers.

## 2. Materials and Methods

In autumn 2020, the cross-sectional study assessing the seroprevalence of the SARS-CoV-2 virus was conducted in three Polish cities located in the Silesian Voivodeship (Gliwice, Katowice, and Sosnowiec). The methodology of this study and some of the results have already been presented and discussed in our previous articles [12,13]. The main research tools were questionnaires and measurement of anti-SARS-CoV-2 immunoglobulins (IgG and IgM). Antibodies were measured against S1 proteins (IgG) and modified nucleocapsid protein (IgM) of SARS-CoV-2 in serum and the results were expressed as ratios (test/control extinction), according to the following scale: ratio < 0.8 = negative result, ratio 0.8–1.09 = questionable result, ratio > 1.09 = positive result. The manufacturer’s log files (EuroImmun Polska Sp. z o.o., Wrocław, Poland) reported a specificity of 99% (IgG) and a maximum sensitivity of 88% (IgG). Independent values included demographic, socioeconomic, health-related characteristics, and type of sampling. However, some important points that must be mentioned to introduce to the current analysis are discussed and repeated below. Initially, the study designed assumed the acquisition of subjects using random sampling method stratified by age and gender. Limited participation was expected at the planning stage of the study and was taken into account when calculating the minimum sample size. As only 1167 people (19.5% initially invited) agreed to participate in the study, supplementary recruitment was introduced resulting in additional 4321 volunteers and a total of 5488 participants. The final sample size met the estimated minimum sample size. Therefore, we decided to compare these two groups of participants to assess the impact of the recruitment method on the results. The first group (1167 participants) included a randomly selected, sex- and age-stratified sample of subjects (which is hereinafter referred to as the random group, RG), recruited using personal invitation letters and postal addresses obtained from a national registry. The second group (4321 volunteers, “VG” group) included those who expressed their willingness to participate in response to an advertisement published in media (including regional newspapers, TV, radio, and Facebook). Moreover, we also provided each primary care physician active in the study area with a complete kit of information on the project. Distributions of age and sex in RG did not differ statistically significantly from analogous distributions in the general population of the Silesian Voivodeship [12].

### 2.1. Statistical Analysis

Both groups (RG and VG) were compared in terms of the results of the questionnaire and serological examinations. The distribution of quantitative variables was initially assessed, with deviations from the normal distribution found (Shapiro–Wilk test). Therefore, between-group differences in the distribution of quantitative variables were analyzed using non-parametric tests (Mann–Whitney test). Analysis involving qualitative variables was performed using chi-square test (or Fisher’s exact test for cells with excepted counts less than 5). The results of descriptive analysis included the relative frequencies (percentages) for categorical variables and the medians with interquartile ranges (IQR) for quantitative variables. Additionally, we deepened the analysis of the determinants of recruitment (binary dependent variable: random recruitment vs. self-selection) applying the multivariate logistic regression model. Only the variables considered statistically significant at the stage of univariate analysis were included in the regression model and, finally, odds ratios (OR) with 95% confidence intervals (CI) were counted. Interpretation of statistical tests was conducted according to the criterion *p* < 0.05. All analyses were performed using the R 4.1.0 statistical environment (2021, R Core Team, GNU General Public License; R Foundation for Statistical Computing, Vienna, Austria).

### 2.2. Ethical Approval

The study was approved by the Ethics Committee of the Medical University of Silesia in Katowice (14 November 2020; the number of approval PCN/0022/KB1/61/20) and was registered at the ClinicalTrials.gov (accessed on 26 March 2021) PRS system with NCT04627623 identifier. The protocol and the course of the study were in line with Helsinki’s declaration. All participants signed informed written consent to participate in the study.

## 3. Results

The RG group consisted of 1167, while the VG group included 4321 subjects. In the first stage, several factors were identified using simple difference tests, including sociodemographic features, differentiating participants depending on the recruitment model. Compared with randomly selected subjects, the volunteers were more often females, they were younger and more often professionally active (Table 1). They more often had a history of contact with a COVID-19 patient (33% vs. 12.8% in RG, *p* < 0.001), post-contact nasopharyngeal swab (17.5% vs. 13%, *p* = 0.001), fewer comorbidities (12.9% vs. 18%, *p* < 0.001; more details in Table 2), as well as declared the occurrence of symptoms that might suggest infection with SARS-CoV-2 (detailed results presented in Table 3).

Additionally, a significantly higher percentage of positive IgG results was found in this group (23.5% compared to 11.4% in RG, *p* < 0.001), denoting their contact with the SARS-CoV-2 virus (Table 4). We did not find any statistically significant differences neither in the frequency of positive IgM tests between the compared groups nor in associations between IgM results and other variables, including symptoms.

Furthermore, most of them were vaccinated against tuberculosis (78.2% vs. 68.7%, *p* < 0.001), but with no differences for the influenza seasonal vaccination. There were no differences in the recruitment structure to the inhabited city, seek for medical help to symptoms, and subject’s body mass index (BMI). Detailed characteristics are presented in Table 1, Table 2, Table 3 and Table 4.

Most of these relationships were confirmed in the multivariable regression model (Table 5), which had a McFadden pseudo R2 index of 0.3 and Cragg and Uhler’s pseudo R2 of 0.4. Based on these results, subjects who were self-selected to the study were less often male (OR = 0.76, 95% CI 0.64–0.89), more often younger (OR = 0.98; 95% 0.98–0.99), recruited in December (OR = 411.58, 95% CI 246.24–727.09), professionally active (OR = 1.92; 95% CI 1.62–2.28), had contact with a COVID-19 case (OR = 1.49; 95% CI 1.15–1.95), and had a higher IgG ratio (OR = 1.08; 95% CI 1.03–1.15). According to the multivariable model, the presence of comorbidities did not significantly affect study participation.

## 4. Discussion

The objective of our study was to compare demographic, socio-economic, health-related characteristics, and the anti-SARS-CoV-2 immunoglobulin IgG occurrence among the randomly selected group and the group composed of volunteers. The major finding of our study was the identification of a selection bias, defined as the difference in many characteristics between randomly selected subjects and volunteers. Our findings not only showed between-group statistically significant differences in the distribution and associations of many important variables, but also allowed us to measure the size of the bias.

As expected, our study confirmed the between-group differences in the distribution of pertinent variables and perhaps the most important difference concerned the frequency of seropositivity with an apparent overestimation of the frequency of SARS-CoV-2 infection in the general population. However, the bias was seen in relation to all relevant aspects of the study: description, analysis differences, and associations.

With regard to the structure of examined groups, our results confirmed a larger participation of women and younger people among volunteers and this finding was consistent with previous observations in this regard [1,14]. This finding was also in line with the results of other cross-sectional studies concerned with different goals of public health, in which females engage more frequently [14,15,16,17]. Importantly, within our study and compared with randomly selected subjects, the volunteers had a significantly higher percentage of positive IgG test results, probably due to their more frequent contact with the SARS-CoV-2 coronavirus. Such an association was suggested by the information provided by the questionnaire. It could not be excluded that the people who had had contact with individuals positively diagnosed with novel coronavirus infections were more likely to use the offered opportunity to check their serological status. Another self-selection mechanism might have resulted from a greater personal interest in the subject of the study, and a higher awareness and perception of the impact of the COVID-19 pandemic. As mentioned above, the volunteer bias shown in our study affected all pertinent aspects of research, starting with the baseline description of the subjects. Such an observation was similar to the results of the New Zealand study in which volunteers and sampled subjects differed significantly, mainly in socio-behavioral respects [15]. In another current publication, the authors concluded that depending on the sampling location and time, people who are present to be sampled may be at a higher or lower risk of COVID-19 than the average risk in the source population [18]. Therefore, our results provided by simple analyses were verified with the use of a multivariable logistic regression model which allowed to identify the factors characterizing the group of volunteers vis-à-vis the randomly selected group. It was established that they included: gender (female), age (younger), employment status (active), history of contact with COVID-19 case (positive), and IgG ratio (higher). These findings were broadly consistent with what has been reported so far, especially with regard to the dominance of women among volunteers [1,2,4].

The results of another recently published study showed that the two different sampling methods had a significant impact on the reporting of COVID-19 symptoms leading to different frequencies (symptomatic subjects: 28.2% in open invitation group and 16.2% in random sample), which was in line with our results. Moreover, the overall prevalence rate of active COVID-19 cases within the open invitation sample (13.3%) was almost twice as big as that found in the random sample (6.9%) [19].

The method of recruiting volunteer participants in our study was similar to the methods used in other population-based studies [9]. In our study, we employed a variety of means in order to reach the population, ranging from traditional press, through television, and ending with the internet. It could be expected that such a “diversified” approach increases the probability of reaching different groups of potential participants in comparison with only one type of information channel.

There is ample evidence that voluntary recruitment may lead to a distorted assessment of the problem under investigation [4,20,21,22]. The issue is far from being resolved particularly in the field of population-based surveys targeting important public health topics. Moreover, survey response rates in cross-sectional studies have been declining for decades [23] and there is a need to develop better methods and practices of surveying in epidemiological studies. Some possibilities arise from the advent of new technologies and the introduction of effective social network support. Such methods should be reviewed and verified in real-life studies with a view of “good epidemiological practice”. The lack of effective population-based recruitment strategies in large surveys affects the reliability of the results and significantly hampers the proper interpretation of the research findings [24,25]. Our study showed that the practical method of such an evaluation may involve a direct comparison of the results obtained using novel recruitment techniques and random sampling, which remain essential components of standard epidemiological studies designs [18,26].

Our study had some limitations. First of all, it was unjustified to claim that our selection procedure of the random sample had resulted in a fully representative sample of the source population in all aspects pertinent to the study objectives. Even if the sample had met the requirements of sex and age distribution and its size had been satisfactory in terms of the desired study power, the participation rate was rather low (19%). However, the problem was general and with several assumptions, it was a random sampling that allowed inferences to a source population. Another issue that potentially hampered the conclusion regarding the exact impact of volunteer bias as analyzed in our study stemmed from a low participation rate in the random-selection phase of the project. The problem was universal and it could not be excluded that a better participation would have resulted in more reliable estimates. However, the findings of our study reflected a real life scenario and, with all potential pitfalls, the results of our comparison between representative and volunteers groups described the presence of a volunteer bias. Moreover, the random selection might have missed the individuals who had been treated for COVID-19 or on quarantine and not responded to the invitation. The methods used to address the objective of our study allowed conclusive real-life comparisons which was the strength of our investigation. Both groups of subjects were inhabitants of the same precisely defined area (three towns; population 600,000) and all subjects were examined by one team using the same methods (questionnaire, IgG test), including one diagnostic laboratory. Moreover, both groups were examined in the same study period (October–November 2020). Additionally, the fear of being infected during the procedures carried out in the study could lead to the exclusion of some potential subjects from the participation (non-responder bias), which may be one of the limitations of the study.

## 5. Conclusions

The findings of the study confirmed that surveys limited to volunteers are biased. The presence of the bias may seriously affect and distort inference and make the generalizability of the results more than questionable. The impact of bias on the external validity of the study depends on its size. Specific outcomes of comparisons performed within our study showed that, with regard to such an important issue as SARS-CoV-2 infection, the much-needed evidence on the description and cause–effect associations unequivocally depend on the recruitment procedure. Effective control over selection bias in surveys, including volunteers, is virtually impossible and its impact on the survey results is impossible to predict. However, whenever possible, such surveys could include a small component of a random sample to assess the presence and potential effects of selection bias.

## Figures and Tables

**Table 1 ijerph-18-09928-t001:** Basic sociodemographic characteristics (relative frequencies presented as percentages or median with interquartile ranges.

		All Subjects n = 5488	Randomized n = 1167	Self-Selected n = 4321	*p*
Quantitative variables (median and IQR)					
Age	Median (IQR)	44 (32–57)	48 (34–63)	43 (31–55)	<0.001
Body mass index	Median (IQR)	25.2 (22.3–28.7)	25.6 (22.5–29.0)	25.2 (22.2–28.5)	0.08
	Missing	6.7%	9.8%	5.9%	
Categorical variables (percentages)					
Town	Gliwice	32.5	33.6	32.2	0.7
	Katowice	34.2	33.8	34.3	
	Sosnowiec	33.3	32.6	33.5	
Gender	Female	58.3	51.3	60.1	<0.001
	Male	41.7	48.7	39.9	
Professional activity	Has no job	31.2	43.9	27.7	<0.001
	Has a job	67.6	55.5	70.9	
	Works and studies	1.2	0.6	1.4	
Month of serological examination	10/2020	9.1	38.2	1.3	<0.001
	11/2020	74.0	60.1	77.7	
	12/2020	16.9	1.7	21.0	
Vaccinated against tuberculosis	Yes	76.2	68.7	78.2	<0.001
Vaccinated against seasonal influenza last year	Yes	16.5	12.2	17.6	<0.001
Medical help following symptoms	Yes	21.6	19.5	22.2	0.2
	Missing	2.5	3.5	2.2	
Contact with a confirmed case of COVID-19	Yes	28.7	12.8	33.0	<0.001
Had a post contact test	Yes	16.6	13.0	17.5	0.001

Legend: n—number of participants; *p*—*p*-value; IQR—interquartile range; IgG—immunoglobulin G; IgM—immunoglobulin M.

**Table 2 ijerph-18-09928-t002:** The burden of comorbidities in the study group in the entire study group and depending on the recruitment mode (percentages).

Comorbidity	All Subjects n = 5479	Randomized n = 1164	Self-Selected n = 4315	*p*
Any comorbidities	14.0	18.0	12.9	<0.001
Coronary artery disease	0.6	1.1	0.4	0.01
Myocardial infarction	0.5	0.9	0.3	0.05
Heart failure	0.3	0.5	0.3	0.2
Valvular heart disease	0.4	0.8	0.3	0.07
Stroke	0.3	0.6	0.2	0.07
Chronic obstructive pulmonary disease	2.0	3.2	1.7	0.003
Asthma	6.7	7.1	6.6	0.6
Chronic Allergic Disease	11.5	9.9	11.9	0.06
Diabetes	6.5	9.7	5.6	<0.001
Cancer	4.2	6.4	3.6	<0.001
Chronic Rheumatological Disease	3.5	5.3	3.0	<0.001
Autoimmune disease	6.7	6.2	6.9	0.4

Legend: n—number of participants; *p*—*p*-value.

**Table 3 ijerph-18-09928-t003:** Results of epidemiological interview, including prevalence of declared symptoms (percentages) with *p*-values of difference tests.

Symptoms	All Subjects n = 5479	Randomized n = 1164	Self-Selected n = 4315	*p*
Fever 38 °C	18.7	15.1	19.7	<0.001
Chills	20.7	14.7	22.3	<0.001
Fatigue	47.9	36.6	51.0	<0.001
Muscle ache (myalgia)	31.5	23.7	33.7	<0.001
Sore throat	35.7	29.3	37.4	<0.001
Cough	36.5	33.0	37.4	0.005
Runny nose (rhinorrhoea)	46.6	43.4	47.5	0.01
Shortness of breath	16.1	13.5	16.8	0.006
Wheezing	7.7	7.0	7.9	0.4
Chest pain	14.3	13.0	14.6	0.2
Headache	44.3	33.6	47.2	<0.001
Conjunctivitis	5.1	4.2	5.4	0.1
Nausea/vomiting	7.9	6.7	8.2	0.09
Abdominal pain	13.7	11.5	14.3	0.01
Diarrhea	15.6	11.9	16.6	<0.001
Loss of smell/taste	13.5	9.2	14.6	<0.001
Other symptoms	6.0	5.0	6.3	0.08

Legend: n—number of participants; *p*—*p*-value.

**Table 4 ijerph-18-09928-t004:** The prevalence of positive serological results and ratios of IgG and IgM: percentages or median values with interquartile ranges (IQR).

Variables	All Subjects n = 5479	Randomized n = 1164	Self-Selected n = 4315	*p*
*IgG*				
% positive	21.0	11.4	23.5	<0.001
ratio-median (IQR)	0.2 (0.1–0.5)	0.2 (0.1–0.3)	0.1 (0.1–0.9)	<0.001
*IgM*				
% positive	5.0	4.8	5.1	0.8
ratio-median (IQR)	0.1 (0.1–0.2)	0.1 (0.1–0.3)	0.1 (0.1–0.2)	0.004

Legend: n—number of participants; *p*—*p*-value; IQR—interquartile range; IgG—immunoglobulin G; IgM—immunoglobulin M.

**Table 5 ijerph-18-09928-t005:** Characteristics of the respondents in univariate analyses and multivariate binary logistics regression model in relation to the recruitment mode as dependent variable.

	Binary Logistic Regression-Dependent Variable: Mode of Recruitment (Random = 0 vs. Self-Selected = 1)					
Explanatory variables	Univariable			Multivariable		
	OR	95% CI	*p*	OR	95% CI	*p*
Age	0.99	0.98 to 0.99	*	0.98	0.98 to 0.99	*
Body mass index	0.99	0.98 to 1.00	NS	X	X	X
IgM ratio	0.99	0.90 to 1.09	NS	X	X	X
IgG ratio	1.20	1.14 to 1.26	*	1.08	1.03 to 1.15	*
Gender male vs. female	0.70	0.61 to 0.80	*	0.76	0.64 to 0.89	*
Comorbidities Yes vs. No	0.68	0.57 to 0.81	*	0.90	0.71 to 1.12	NS
Month of serological examination						
*November vs. October*	38.09	28.75 to 51.35	*	35.52	26.48 to 48.78	*
*December vs. October*	361.41	218.45 to 624.13	*	411.58	246.24 to 727.09	*
Professional activity						
*Has a job vs. has no job*	2.01	1.77 to 2.31	*	1.92	1.62 to 2.28	*
*Works and studies vs. has no job*	3.60	1.75 to 8.70	*	2.53	1.13 to 6.80	0.039
Contact with a confirmed COVID-19 case						
*No vs. do not know*	0.61	0.52 to 0.73	*	0.67	0.55 to 0.83	*
*Yes vs. do not know*	2.36	1.89 to 2.95	*	1.49	1.15 to 1.95	0.002
Post-contact test						
*No vs. do not know*	0.98	0.60 to 1.54	NS	X	X	X
*Yes vs. do not know*	1.39	0.84 to 2.26	NS	X	X	X
Vaccinated against tuberculosis						
*No vs. do not know*	0.79	0.60 to 1.03	0.08	0.98	0.69 to 1.40	NS
*Yes vs. do not know*	1.54	1.31 to 1.80	*	1.43	1.18 to 1.75	*
Vaccinated against seasonalinfluenza last year						
*No vs. do not know*	1.16	0.38 to 3.00	NS	X	X	X
*Yes vs. do not know*	1.79	0.57 to 4.69	NS	X	X	X

Legend: OR—odds ratio; CI—confidence interval; *p*—level of significance; *—*p*-value ≤ 0.001; NS—*p*-value > 0.1; X—not included in the multivariable model.

## Data Availability

Data are available for research upon approval from the Medical Research Agency, Poland. To obtain data, researchers need to submit an analysis proposal to the corresponding author for evaluation and processing the request to the Medical Research Agency, Poland.

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
