# Peer review of "Impact of Two Different Recruitment Procedures (Random vs. Volunteer Selection) on the Results of Seroepidemiological Study (SARS-CoV-2)"

_ijerph, 2021, doi:10.3390/ijerph18189928_

Round 1

Reviewer 1 Report

the article has clear, easy-to-understand language.
Although it uses a current event, COVID-19, it does not bring anything
innovative or original.
However, I believe that it contributes to support
improvements in the recruitment of populations for future studies
It should be revised for a better discussion of the results.  

Reviewer 2 Report

This is a short report comparing participant characteristics and anti SARS-CoV-2 spike IgG positivity from two sample populations at the end of 2020. The report shows that self-volunteered individuals recruited through media advertisements were biased both in their demographics and their serostatus, highlighting limitations of non-random recruitment in accurately tracking seroprevalence. Overall, the study is fairly well described and adequately supports its claims with data and appropriate statistics.

  • My broad concern is that the treatment of the seroepidemiological results is extremely brief. The only serology data we are shown is the overall percentage who tested positive by IgG to SARS-CoV-2 spike. I gather that the random group serology is well described in another publication, but is the volunteer group serology described elsewhere? Many questions come to mind – how do the titer distributions differ? Are there associations between seropositivity and eg. reported symptoms in the volunteer group? Given that this paper concerns the effect of recruitment bias on SARS-CoV-2 seroepidemiology, I found the treatment of the serological data far too brief. The IgM results are not reported at all.
  • Related, I understand that the topic of this paper is recruitment bias, but it is surprising to not see any attempts to adjust/reweight the volunteer group serology results, as has been done in other seroepidemiological studies with biased recruitment. The authors state in the abstract that “whenever possible such surveys could include a small component of a random sample to assess the presence and potential effects of selection bias”, but then do not attempt to provide any insights on SARS-CoV-2 seroprevalence using the non-random group data.
  • The introduction is also very brief, with just a few references on the general topic of volunteer bias in survey recruitment. I would suggest citing some more SARS-CoV-2 seroepidemiology studies to provide more context here. Can you provide some example studies that report seroprevalence from participants recruited not-at-random (please cite more than just this, but one eg. Routledge et al. Nature Communications 2021 https://doi.org/10.1038/s41467-021-23651-6)? Are there any other seroprevalence studies in Poland?
  • Some more information on the volunteer recruitment would be helpful. The only information we have is that “the second group included those who expressed their willingness to participate in response to an advertisement published in media (Facebook).” Was Facebook the only media platform used? What did the advertisement say? How were these ads deployed and targeted? Were there any criteria to recruit from those who “expressed a willingness to participate”?
  • I would like to see some discussion about the generalizability of these recruitment biases. The authors double down on the bias towards females in volunteer-based recruitment, but it is not clear if this is a reflection of volunteering in general or (more likely) dependent on the advertisement platform.
  • Typo L156: SAS-CoV-2; Typo L178 “simple analyzes”
  • L181: shouldn’t this be “gender (female)”?

Reviewer 3 Report

This study set a noble goal of identifying the biases exposed when comparing COVAR2 seropositivity in representative and volunteer samples. However, the Methods explain that the sample that was selected to be regionally representative had barely a 20% capture rate, calling into question its designation as unbiased. Without a solid picture of the true situation, the differences reported between the volunteers in the sample designed to be representative and the volunteers in the convenience sample are not interpretable. While the Authors mention this as a limitation, I find it to be a fatal flaw.

Round 2

Reviewer 3 Report

While I too am resigned to the necessity of making the best of biased data for clinical use, this report is specifically oriented to the methodological question of identifying biases introduced by a volunteer sample. This requires a representative 'gold standard' sample, and the sample selected to representative, while matched on sex and age, was, unfortunately,  too small to ensure control of other confounding factors.  I do not believe that these data can be used to accomplish the Authors' stated objectives.

Author Response

Dear Reviewer, Please find our response below.

Reviewer's comment:

"While I too am resigned to the necessity of making the best of biased data for clinical use, this report is specifically oriented to the methodological question of identifying biases introduced by a volunteer sample. This requires a representative 'gold standard' sample, and the sample selected to representative, while matched on sex and age, was, unfortunately,  too small to ensure control of other confounding factors.  I do not believe that these data can be used to accomplish the Authors' stated objectives."

Authors' response:

It is a pity that the Reviewer has such an opinion, but  they have a holy right to do so. We cannot agree with the opinion that cross-sectional studies (which were  the type of our study) require a gold standard (which is typical for case-control studies). We have ensured the only possible method of control of the representativeness of the studied sample, about which we had written several times in our other publications regarding the frequency of SARS-CoV-2 infections amongst the studied agglomeration [1-4]. No one has yet questioned the accuracy of our cross-sectional study until now, where in the case of a methodological paper in which we have criticized two separate recruitment methods (random sample vs. volunteers) within the conducted and completed research, such comments were made, and we perceive that as unfair.   References: 1. Barański Kamil, Brożek Grzegorz, Kowalska Małgorzata, Kaleta-Pilarska Angelina, Zejda Jan Eugeniusz. Impact of COVID-19 pandemic on total mortality in Poland. Int.J.Environ.Res.Publ.Health 2021 : Vol.18, No.8, p.1-8
2. Zejda Jan E., Brożek Grzegorz M., Kowalska Małgorzata, Barański Kamil, Kaleta-Pilarska Angelina, Nowakowski Artur, Xia Yuchen, Buszman Paweł. Seroprevalence of Anti-SARS-CoV-2 antibodies in a random sample of inhabitants of the Katowice Region, Poland. Int.J.Environ.Res.Publ.Health 2021 : Vol.18, No.6, p.1-11
3. Kowalska Małgorzata, Barański Kamil, Brożek Grzegorz, Kaleta-Pilarska Angelina, Zejda Jan E. COVID-19 related risk of in-hospital death in Silesia, Poland. Pol.Arch.Med.Wewn. 2021 : Vol.131, No.4, p.339-344
4. Kowalska Małgorzata, Niewiadomska Ewa, Barański Kamil, Kaleta-Pilarska Angelina, Brożek Grzegorz, Zejda Jan Eugeniusz. Association between Influenza Vaccination and Positive SARS-CoV-2 IgG and IgM Tests in the General Population of Katowice Region, Poland. Vaccines 2021 : Vol.9, No.5, p.1-9, id. art. 415